# Physical Activity Level and Perspectives of Participants Transitioning from Onsite to Virtual Cardiac Rehabilitation during the Early COVID-19 Pandemic: A Mixed-Method Study

**DOI:** 10.3390/jcm11164838

**Published:** 2022-08-18

**Authors:** Lais Manata Vanzella, Gabriela Lima de Melo Ghisi, Tracey Jacqueline Fitchett Colella, Jillian Larkin, Luiz Carlos Marques Vanderlei, Susan Marzolini, Scott Thomas, Paul Oh

**Affiliations:** 1Toronto Rehabilitation Institute, University Health Network, Toronto, ON M4G 2V6, Canada; 2Lawrence S. Bloomberg Faculty of Nursing, University of Toronto, Toronto, ON M5T 1P8, Canada; 3Faculty of Kinesiology and Physical Education, University of Toronto, Toronto, ON M5G 2W6, Canada; 4School of Technology and Sciences, São Paulo State University, Presidente Prudente 19060-900, SP, Brazil

**Keywords:** cardiac rehabilitation, COVID-19, physical activity, mixed methods

## Abstract

This mixed-method study aimed to compare physical activity (PA) patterns of a cross-over cardiac rehabilitation (CR) cohort with a center-based CR cohort and to explore barriers and facilitators of participants transitioning and engaging in virtual CR. It included the retrospective self-reported PA of a cross-over CR cohort (n = 75) and a matched center-based CR cohort (n = 75). Some of the participants included in the cross-over cohort (n = 12) attended semi-structured focus group sessions and results were interpreted in the context of the PRECEDE-PROCEED model. Differences between groups were not observed (*p* > 0.05). The center-based CR cohort increased exercise frequency (*p* = 0.002), duration (*p* = 0.007), and MET/minutes (*p* = 0.007) over time. The cross-over cohort increased exercise duration (*p* = 0.04) with no significant change in any other parameters. Analysis from focus groups revealed six overarching themes classified under predisposing factors (knowledge), enabling factors (external support, COVID-19 restrictions, mental health, personal reasons/preferences), and reinforcing factors (recommendations). These findings suggest an improvement of the PA levels of center-based CR cohort participants pre-pandemic and mitigated improvement in those who transitioned to a virtual CR early in the pandemic. Improving patients’ exercise-related knowledge, provider endorsements, and the implementation of group videoconferencing sessions could help overcome barriers to participation in virtual CR.

## 1. Introduction

The Coronavirus disease 2019 (COVID-19) pandemic has affected more than 400 million people and caused more than 5 million deaths worldwide [1]. The prevalence of and mortality from COVID-19 is high in people living with cardiovascular disease (CVD), especially those with coronary artery disease and congestive heart failure [2,3,4]. Studies have suggested that one of the long-term complication of COVID-19 is the development of CVD [5,6]. Investments in effective strategies for CVD treatment and prevention are proven to be crucial, both during and beyond the COVID-19 pandemic [7,8]. However, protective measures implemented to help contain the spread of the virus (i.e., mask mandate, social distancing, staying home, and closing businesses, services, and public spaces) [9] have directly influenced people’s lifestyles, and challenged how people living with CVD are able to routinely and confidently access essential healthcare services, including cardiac rehabilitation (CR) [10].

CR is a cost-effective comprehensive program that includes medical evaluation, exercise prescription, psychosocial support, patient education, and counseling for cardiac risk factor modification and strategies to promote behavior change [11,12]. Benefits associated with participation in CR programs include reduction in all-cause mortality, re-hospitalization, and cardiovascular morbidity [13,14], as well as improvement in mental health, quality of life [14,15], and exercise capacity [16,17]. Although benefits are well-known, many CR programs across Canada [18] and worldwide [19] have closed or shifted to home delivery due to the COVID-19 pandemic.

Home-based CR programs were initially designed to help increase participation in CR programs by overcoming access-related barriers, such as geographic and logistical barriers [20]. Recent systematic reviews have shown that center-based and home-based CR have equivalent effects on muscle strength [21], functional capacity [21,22,23], quality of life [22,23], physical activity behavior [23], medication adherence, smoking behaviour, depression, and cardiac-related hospitalizations [23,24,25]. Although considered an alternative to the delivery of CR care, home-based program models were only offered by 285 (31.1%) programs in 51 countries before the pandemic [20].

During the pandemic, results from a pan-Canadian survey of CR programs revealed that 58.7% of CR programs that remained open had transitioned to a home-based delivery model [18]. Home-based CR with virtual supervision (telephone, videoconferencing communication, email, text, or other messaging solutions, smartphone applications, online platforms, and wearable devices) was viewed as a great opportunity to continue providing care for those with CVD while respecting health protection restrictions [26]. Overall, advantages of the delivery of virtual CR compared to center-based models include higher levels of patients satisfaction [27], easy access [28], and lower cost [29]. Conversely, barriers to implementation and participation in virtual CR include lack of technology, infrastructure, and financial sustainability [30,31,32].

The unique situation caused by the restrictions adopted to contain the spread of the COVID-19 virus created particular challenges to participation in CR programs, including the interruption of supervised on-site CR, changes to the mode of delivery without sufficient opportunities for planning and training, employee redeployment [18,19], a lack of equipment necessary to exercise at home, restrictions to patient access, and patient difficulties regarding the use of technology [18]. Therefore, this mixed-method study aimed to: (1) compare physical activity patterns of participants in a cross-over CR cohort who transitioned from a center-based to a virtual CR program during the COVID-19 pandemic with those who initiated and completed their participation in a center-based CR program before the pandemic; and (2) to explore barriers and facilitators to transitioning and engaging in virtual CR model.

## 2. Methods

### 2.1. Study Design

This was a mixed-method study that evaluated retrospective and qualitative data of individuals participating in a comprehensive CR program at the Toronto Rehabilitation Institute, University Health Network, ON, Canada. The study was approved by the Institutional Research Ethics Board (20-5612.0 and 20-5759).

Figure 1 shows an overview of the sequential explanatory study design consisting of two phases: a retrospective quantitative and a prospective qualitative phase.

The qualitative phase aimed to explain or enhance the quantitative results. Retrospective physical activity data were extracted from the CR clinical record system. Our primary analysis examined and compared retrospective physical activity patterns (exercise frequency, duration, and MET/minutes) reported by individuals participating in a center-based CR one month before the pandemic (1st time point: February 2020) with physical activity patterns reported by this population one month after they transitioned to a virtual CR program early in the pandemic (2nd time point: April 2020). Participants who transitioned from a center-based to a virtual CR program due to the pandemic were identified as a “cross-over CR cohort”. Additionally, we compared physical activity patterns of the cross-over CR cohort to a matched cohort who had completed participation in a center-based CR program in 2019 (1st and 2nd time point were of equal duration in both cohorts). The participant-matching process was conducted in order to mitigate possible selection bias, given the difference in the proportion of dropouts between the original cohorts, which may have left a selected sample. Each individual who participated in the cross-over CR was matched with an individual who participated in a center-based CR model. The matching variables were age, sex, referral diagnosis, and duration of participation in the CR (i.e., attended the same number of pre-scheduled CR sessions). Matched individuals could not be used more than once, which means that a matched individual from a center-based CR was paired with a unique individual from the cross-over CR model. If a match with an individual of the exact same age was not possible, we identified the closest person in age within the same 2-years age group, but with the same sex, referral diagnosis, and duration of participation in the CR. Individuals from whom all match variables’ match-person could not be identified, were excluded. Our secondary analysis examined and compared physical activity patterns of the cross-over and center-based CR cohort considering sex and age (<65 years old vs. ≥65 years old).

A total of 30 individuals included in the early pandemic CR cohort were invited to participate in virtual focus group sessions. Those sessions were designed to explore barriers and facilitators to participate in a virtual CR program during the pandemic.

### 2.2. Participants

This study included males and females greater than 18 years of age that participated in either a center-based CR program or in a cross-over CR program. Participants with incomplete exercise data (i.e., missing duration or frequency of the weekly performed exercise) from clinical records were excluded from the retrospective data analyses and focus group sessions. The center-based CR cohort were not included in the qualitative focus group session, as well as participants from cross-over CR that did not speak and read English.

### 2.3. Setting

Pre-pandemic Center-Based CR: This model was a 26-week program that included 22 weekly group-based supervised and 4 weekly unsupervised (completed at home or in the community) exercise sessions, including range-of-motion exercises, cardiovascular warm-up, aerobic training, resistance training, and a cool down. Participants were followed by a CR exercise provider (case manager) from the beginning to the end of their participation in the CR program. As described elsewhere [33], the goal of aerobic training was to progress patients to 60 min of aerobic exercise, 5 times per week, at a moderate intensity determined by the HR achieved at the anaerobic threshold. Resistance training was introduced during the eighth week of the program, consisting of 10 exercises targeting all major muscle groups. Patients were instructed to keep a detailed record of their aerobic (distance walked/biked, duration, resting and peak heart rate, and rating of perceived exertion) and resistance training sessions (amount of weight lifted, the number of repetitions completed, sets performed, and rating of perceived exertion during each exercise). In addition to supervised exercise sessions, patients received weekly lifestyle education group sessions, 30 min of duration, delivered by a multidisciplinary team and covering different topics including how the heart works, heart health behaviours (exercise, diet), psychosocial health, and self-management [34]. Participants were also offered the opportunity to communicate with a psychologist, social worker, and dietitian to receive specific information about mental health, social support, and healthy eating habits.

Virtual CR: When the pandemic started, participants transitioned from a center-based to a virtual CR model (cross-over CR) to complete their 26 weeks of participation in the CR safely at home. Participants continued to be followed by a CR provider via telephone and email, and received information about their exercise progression and safety, as well as motivational support to overcome barriers to exercise. However, the CR provider might have changed during the transition from a center-based to a virtual CR model due to staff redeployment. The aerobic exercise prescription remained the same as described in the center-based CR program model. The resistance exercise prescription was adapted to home exercises following the Cardiac College Website (Resistance Training | Cardiac College (healtheuniversity.ca, accessed on 19 July 2022). Participants were also offered the opportunity to communicate with a psychologist, social worker, and dietitian. In regards to the educational component, participants were offered the opportunity to engage in educational webinars via online platform available at the Cardiac College Website (Learn Online | Cardiac College (https://www.healtheuniversity.ca/EN/CardiacCollege/Pages/learn-online.aspx, accessed on 19 July 2022). These sessions were delivered on weekdays (Monday to Thursday) and covered topics related to exercise prescription, psychological well being, healthy eating habits, and special topics involving novel information about medications, CVD, and COVID-19.

### 2.4. Quantitative Data: Retrospective Data Extraction

Sociodemographic characteristics (age, sex), duration of participation in the CR program, medication prescribed, referral diagnosis and procedure, and body mass index (BMI), were extracted from the CR clinical records for sample characterization. The physical activity patterns extracted and analyzed in this study included exercise duration, frequency, and METs/minute. Exercise duration was the average minutes spent doing physical activity per week; exercise frequency was the number of days in which physical activity was performed per week. Physical activity duration was converted to MET-minute/week using the corresponding MET values from the compendium of physical activities [35]. An average of two weeks of exercise frequency, duration, and METs/minute reported by participants on the 1st and 2nd time point were considered for analysis in both groups.

### 2.5. Qualitative Data: Semi-Structured Focus Groups

A semi-structured focus group interview guide was developed by the research team based on their expertise, existing literature [34], and the PRECEDE-PROCEDE model [35]. This is an eight-phase framework used by health care providers to determine, develop, implement, and evaluate health programs. Elements of the PRECEDE-PROCEED model that were particularly relevant for this study include the process of evaluation (phase 7), as it describes predisposing, enabling, and reinforcing factors that influence participation in health programs and behaviour change [36]. Appendix A presents an script with questions utilized to guide the focus group sessions. 

Qualitative data were collected from 90-min semi-structured focus group sessions facilitated by study members (LV and JL). Focus group sessions were conducted virtually by videoconferencing through the Microsoft Teams application. Participants were encouraged to identify barriers and facilitators to exercise as part of their participation in virtual CR during the COVID-19 pandemic. Sessions were digitally audio recorded via the Microsoft Teams application and transcribed verbatim (LV), which provided immersion in the data analysis. Field notes were compiled and summarized by JL to facilitate member checking, as participants were given the opportunity to review and clarify the summary of responses at the end of the focus groups session.

### 2.6. Data Analysis

For analysis between two groups (center-based CR cohort and cross-over CR cohort), factors (data analysis “1st time point” and “2nd time point”), and group vs. factors interaction, repeated measures ANOVA was used. Mauckly’s test was used to check the sphericity violation and Greenhouse–Geisser correction was used when violation was assumed. Sub-analysis of the pandemic CR cohort stratified by age (<65 and ≥65 years old), and sex were also conducted using a repeated measures ANOVA test to compare factors (data analysis “1st time point” and “2nd time point”), groups (according to the above stratification of age, sex), and group vs factor interaction. Data were not normally distributed and were log transformed for the analysis. SPSS version 22.0 was used for all data analysis. The statistical significance was set as 5%.

For the qualitative data, focus group sessions were conducted until the saturation point was reached. The concept of saturation has become the gold standard by which sample sizes for research qualitative inquiry, including focus groups, are determined [37,38]. Data saturation was achieved when no new themes emerged from thematic analysis following the third focus group session, after interviewing a total of 12 individuals. The transcripts were reviewed by study members (LV, GLMG) and the initial systematic coding and categorization of the transcripts were conducted through repeated reading and line-by-line analysis of the text by two independent coders (LV, GLMG). Braun and Clark’s [39] six-phased, iterative, and reflexive approach to thematic analysis was used to guide the identification, analysis, and reporting of themes emerging from the focus groups sessions within the context of the PRECEDE-PROCEED model. Thematic content analysis includes: (1) Familiarizing with the data: researchers engaged in a ‘repeated reading’ of the data in an active way, in order to identify meanings and patterns. During the familiarization and transcription process, the researchers took notes by generating initial ideas about the information emerging from the data; (2) Generating initial codes: codes were generated by two independent researchers as a basic statement that reflected information of the transcribed data that appears to be interesting for analysis, in a meaningful way. Codes were identified, matched with the data extracted, and grouped according to repeated patterns and overarching potential themes. Consensus was achieved through discussion and debate between the team members; (3) Searching for themes: all relevant coded data extracted were identified at a broader level of themes; (4) Reviewing themes: emerging themes were generated and reviewed by the team to ensure that they reflected the dataset and to allow for coding any additional data for themes that were missed in the earlier coding stages; (5) Defining and naming themes: researchers “defined and refined” the emerged themes, allowing for the identification of the overall themes and the determination of what aspects of the data were captured by each theme. Data saturation was achieved when no new theme emerged from thematic analysis; (6) Producing the report: data were reported using statements that captured the essence of the emerged themes [36,39]. Trustworthiness of findings was facilitated through team consensus and credibility by the well-established criteria for reporting qualitative research [40].

## 3. Results

Figure 2 shows the study flow diagram. For the retrospective data, a total of 1559 individuals were assessed for eligibility. From those individuals, 1409 were not included in the analysis, 576 from the cross-over CR program, and 833 from the the center-based CR program. Data from 150 individuals were analyzed. For the qualitative data, a sample of 30 random individuals from the cross-over CR cohort included in the retrospective portion of this study were invited to participate in the focus groups session. A total of 12 individuals accepted and were included in the analysis. Data saturation was achieved following the third focus group session, as this session produced no new information and no new themes emerged from the thematic analysis.

### 3.1. Retrospective Data

A total of 150 participants were included in the analysis (cross-over CR cohort, n = 75; center-based CR cohort, n = 75). Table 1 shows the characterization of these participants at entry to CR.

There were significant differences between BMI (center-based CR cohort = 28.3 ± 4.5 vs. cross-over CR cohort = 26.6 ± 3.5; *p* = 0.036) and antidiabetic agent medication (center-based CR cohort = 7, 9.3% vs. cross-over CR cohort = 20, 26.7%; *p* = 0.006). There were no significant differences in other variables (*p* > 0.05).

The mean age of the center-based CR cohort was 67.1 ± 10.0 years vs. 66.9 ± 10.3 years for the pandemic CR cohort and center-based CR cohort. The most prevalent referral diagnosis/procedures in both groups were coronary artery disease—coronary angioplasty and CABG (center-based and pandemic CR cohort, 58.7% and 17.3% respectively).

Table 2 shows the comparison between the physical activity patterns of the center-based CR and the cross-over CR cohorts.

There were no significant differences between groups (center-based and the cross-over CR cohort; *p* > 0.05) and the factor vs. group interaction (*p* > 0.05) for any of the physical activity patterns analyzed. However, differences between factors (1st time point vs. 2nd time point) were observed (exercise frequency: *p* = 0.022; Partial Eta square = 0.04; exercise duration: *p* = 0.001; Partial Eta square = 0.07; METs/minute *p* = 0.001; Partial Eta square = 0.07). The center-based CR cohort significantly improved the exercise frequency (*p* = 0.002) and duration (*p* = 0.009) as well as the METs/minute (*p* = 0.007) from the 1st to the 2nd time points. Significant improvement from the 1st to the 2nd time point in the cross-over CR cohort was observed only for exercise duration (*p* = 0.041).

Table 3 shows a sub-analysis comparing physical activity data of the cross-over CR cohort considering sex and age.

When divided by age (<65 years old vs. ≥65 years old), exercise frequency was significantly different between factors (*p* = 0.035; Partial Eta Square = 0.031), groups (*p* = 0.002; Partial Eta Square = 0.095), and factors vs. group interaction (*p* = 0.030; Partial Eta Square = 0.061). The pandemic cohort <65 years old significantly improved the exercise frequency from the 1st to the 2nd time point (*p* = 0.011). Differences between groups reveled that the exercise frequency was, in average, lower in participants aged ≥65 years old participating in both cross-over and center-based CR programs (cross-over CR cohort <65 years old vs. ≥65 years: *p* = 0.018; cross-over CR cohort <65 years old vs. center-based cohort ≥65 years: *p* = 0.033). Differences between factors were also observed for the exercise duration (*p* = 0.028; Partial Eta Square = 0.033). The center-based CR cohort >65 years old significantly improved the exercise frequency from the 1st to the 2nd time point (*p* = 0.031).

When stratified by sex, no differences between groups (*p* > 0.05) or factor vs. group interaction (*p* > 0.05) were observed. However, there were differences between factors for exercise frequency (*p* = 0.006; Partial Eta square = 0.052), exercise duration (*p* = 0.003; Partial Eta square = 0.021), and METs/minute (*p* = 0.015; Partial Eta square = 0.040). Males participating in the center-based CR cohort improved their exercise frequency and duration from the 1st to the 2nd time point (*p* = 0.008 and *p* = 0.026, respectively). Males participating in the cross-over CR cohort improved their exercise duration as well as the METs/minute from the 1st to the 2nd time point (*p* = 0.027; *p* = 0.049, respectively).

### 3.2. Qualitative Analysis

A total of 12 participants (7 females, 58.3%) participated in one of three focus group sessions (Session 1. n = 4, 1 female—33.3%; Session 2. n = 4, 2 female—50.0%; Session 3. n = 4, 3 female—75%). Participants presented different cardiovascular disease diagnoses, including cardiomyopathy (n = 3; 25.0%), coronary angioplasty (n = 3; 25.0%), valve disease (n = 4; 33.3%), and coronary artery bypass graft surgery (n = 2; 16.6%). The mean age of participants was 75.5 ± 8.4 years old and patients had participated in the CR program for a mean of 2.8 ± 1.3 months before they transitioned from onsite to virtual CR during the first wave of the COVID-19 pandemic.

Six main themes emerged following analysis of the focus group sessions. As described, these themes were based on Phase 7 of the PRECEED-PROCEED model, and identified as predisposing, enabling, and reinforcing factors that influenced physical activity in a virtual CR program offered during the COVID-19 pandemic. Figure 3 shows the summary of the focus groups themes and sub-themes.

### 3.3. Predisposing Factors

Predisposing factors broadly refer to everything that might predispose an individual to exercise [37]. A theme that emerged from the focus group sessions, which was classified under predisposing factors, was knowledge. Some participants expressed that their knowledge about the importance of exercise for health motivated them to continue participating in the CR program when it shifted to a virtual model during the COVID-19 pandemic.

“*[…] I was reading a lot of studies and exercise is important for me to have a long life. We try to understand that it is part of that program to keep me healthy...*”. (L, 68yo male)

One participant also stated about the importance of maintaining health habits for a better COVID-19 recovery.

“*[…] the healthier you are as a person, the better chance you have of recovery if you do get COVID*”. (A, 74yo female).

### 3.4. Enabling Factors

Enabling factors are those that make it possible for individuals to change their behaviours or their environment, including resources, social support, and skills. Themes that emerged from the focus group sessions, which were classified under enabling factors, were the following: external support, COVID-19 restrictions, mental health, and personal reasons/preferences. Participants reported many factors that acted as barriers and facilitators of their ability to exercise while participating in a virtual CR model offered during the COVID-19 pandemic. Some participants experienced a loss of connection with CR providers, which was associated with staff redeployment, and a lack of in-person monitoring.

“*[…] I certainly didn’t feel a connection with the new rehab coordinator (CR supervisor)*”. (S, 74yo female)

“*You know, like, you felt that there was somebody following you? were, when the rehab closes. I felt like I was on my own*”. (E, 72yo female)

On the other hand, some participants felt supported and motivated by their family members and CR providers to continue exercising during the pandemic. That facilitated the continuity of their participation in the CR after the transition from a center-based to a virtual program model.

“*[…] I’ve got my husband’s an exercise maniac. So yeah, he’s an inspiration*”. (V, 75yo female) “*[…] I would like to give my compliment to whoever works in Rehab because the service encourage service, the attention, really, the type of attention you get is unbelievable*”. (G, 88yo male)

The COVID-19 restrictions were also considered a barrier and a facilitator to exercise. Some participants felt that it was difficult to maintain social distancing while exercising outside and to exercise without in-person supervision.

“*Folks don’t seem to realize that older folks are vulnerable*”. (A, 74yo female)

“*[…] I don’t even know if it’s actually benefiting me because I don’t know if I am actually doing it properly or not need that instructor to see what you’re doing*”. (J, 67yo female)

On the other hand, the pandemic motivated participants to increase their physical activity levels, as they had more time to exercise due to the COVID-19 restrictions.

“*I do more exercise during the day because there’s nothing else to distract you from occasionally*”. (C, 81yo male)

Participants also stated the lack of any physical structures needed for exercise was a barrier associated with the COVID-19 restrictions. That included a lack of equipment and a lack of access to exercise facilities due to the COVID-19 restrictions.

“*…the fact that initially they just stopped most the gym and it was during the winter. So I would not go to walk in a mall or something like this. I tried to walk in the corridor...*” (G, 88yo male)

Mental health was another theme that emerged as both barrier and facilitator to exercise. While COVID-19 raised many negative feelings in some of the CR participants, others were able to maintain their positive attitudes, continue with their daily activities, and stay motivated by the desire to keep living and feel alive.

“*I don’t know I’m just very nervous about this is I thought everything was going back to normal and now it’s getting worse again*”. (J, 67yo female)

“*I don’t think about that and it hasn’t really bothered me so to me it’s just the normal everyday doing the same thing based on what I learned at the rehab center*”. (J, 81yo male)

“*Living, it is pretty strong motivation; it is a motivation every single day*”. (C, 61yo female)

Regarding personal reasons and preferences for exercise, some participants identified that it was difficult to learn how to exercise virtually, while others indicated that it was difficult to make time to exercise when they had many family and personal responsibilities.

“*…it went downhill as soon as it went online. It is probably because it was theory, online, and I’m not theory. So even to figure out how far I was walking, doing it by time, and that wasn’t satisfactory, if you want me to do it by kinda what it needs to do on my own. And I had no way of figuring that out*”. (E, 72yo female)

“*I’m always busy doing something and I don’t take as much time as I should to do exercises, I’ll be honest*”. (E, 72yo female)

### 3.5. Reinforcing Factors

Reinforcing factors occur during the start of a behaviour and include incentives and rewards with the increased probability that the behaviour will recur at the next opportunity [37]. In this context, themes that emerged as reinforcing factors included recommendations that were made based on individuals’ personal experiences. Participants indicated they would like to receive a reduced amount of information in the virtual sessions, and would like to have more active (i.e., where participants exercise virtually together) instead of didactic sessions (i.e., where they are taught how to exercise).

“*So it was like I was watching a lot of the online stuff. I just wish that there was a little more exercising near the end… there got to be a little bit too much (informative sessions) like I found a lot of it was repetitive*”. (J, 67yo female)

Specific recommendations raised by participants during the focus groups included the following: (1) Implementation of sessions aiming to better prepare participants on how to exercise at home, (2) Increase the amount of time with CR supervisors, (3) Avoidance of constant changes in CR personnel, and (4) Implementation of virtual interactive group exercise sessions.

“*…suggest having more time, less staff changes, more team spirit, and cover how to do the exercises from home, so that you’re prepared in the event of a major shutdown” (E, 72yo female) “I would say zooming, you know […] doing more sessions with more with people*”. (J, 67yo female)

## 4. Discussion

To our knowledge, this is the first study to investigate the physical activity patterns of individuals who transitioned from in-person to a virtual CR model due to the pandemic, and to compare individuals who participated in a center-based CR model pre-pandemic. The results of this study demonstrate that weekly frequency, duration, and METS/minute were not different between center-based CR and cross-over CR in the first time point (cross-over CR cohort—February 2020; center-based CR cohort—same time of participation in the CR program identified in the cross-over CR cohort). The duration of the exercise significantly increased in both cohorts two months after the first time point. Though frequency and METS/ minute significantly increased only in the center-based CR two months after the first time point, group vs time interaction analysis were not significant, which indicates that the effects of the CR programs were similar in both CR program models.

Overall analysis showed that the cross-over CR cohort maintained or improved some aspects of their physical activity level during their participation in a virtual CR model during the pandemic. However, barriers and facilitators to continue exercising during the COVID-19 pandemic were identified. This information may provide guidance on how CR programs can better address these barriers and improve the quality of their services. Overall, participants who transitioned from in-person to a virtual CR model due to the pandemic identified difficulties with exercising due to the lack of access to equipment, the closure of exercise facilities, and the need to maintain social distancing. Some participants reported that it was difficult to align their exercise routines with their family and work commitments. Conversely, other participants indicated they had more available time to exercise due to the COVID-19 restrictions. Participants had mixed feelings about the pandemic, had different personal learning preferences, missed in-person supervision, and needed support from the CR providers to fully engage in virtual programming. Having knowledge about the importance of exercise motivated participants to continue exercising during the pandemic. Participants recommended that virtual CR programs: implement preparatory sessions on how to exercise at home, increase the number of virtual exercise group sessions (i.e., where participants exercise virtually together), increase the duration of the sessions with the CR supervisors, avoid constant changes in the CR personnel, and implement virtual interactive group exercise sessions (e.g., group videoconferencing).

The literature pertaining to exercise patterns in the overall population during the COVID-19 pandemic is divergent. Lesser et al. identified that 40.5% of physically inactive Canadians reported less exercise performance, while 40.3% of physically active Canadians reported more exercise performance since COVID-19 [41]. Among adults living in Belgium, a large-scale cross-sectional study showed that 36% of highly active people and 58% of lesser active people exercised more during the lockdown compared to periods before the pandemic [42]. Conversely, McCarthy et al. identified a 37% reduction in weekly minutes of exercise after the first full week of lockdown in the United Kingdom [43]. Decreases of physical activity were also observed in specific populations during the COVID-19 pandemic, including among those with chronic diseases [44], the elderly [45], and women [46].

Our results revealed that participants who transitioned from center-based to virtual CR due to the COVID-19 pandemic maintained their exercise duration and METs/minutes, and increased the frequency of their exercise. Those who participated in a center-based CR significantly improved their physical activity patterns. Although improvements in the physical activity patterns were not observed in the cross-over CR cohort, participants were able to continue exercising and maintaining their physical activity patterns. As described elsewhere [45,47], participation in virtual programs using online videos and phones helped individuals improve their physical activity patterns during the pandemic. In our study, participation in a virtual CR model helped individuals maintain their physical activity. Exercise maintenance is an important lifestyle factor that may help avoid the increasing mortality and cardiovascular risk posed by physical inactivity, especially during the pandemic [46].

Additionally, some facilitators of continuing exercise during the pandemic were reported by participants in this study, which included an understanding of the importance of exercise for improving health. As reported in other studies, knowledge regarding exercise and its health benefits is crucial for improving self-management and for increasing the confidence of individuals participating in CR programs [48,49]. Understanding the importance of an exercise routine helps motivate CR participants and the overall population. Such an understanding influences highly active people to continue exercising and lesser active people to be more active [42,50]. The importance of knowledge regarding exercise highlights the need for the availability of comprehensive and evidence-based resources for patients while they participate in virtual CR models. An example is the resource provided by Cardiac College^TM^ (Welcome to Cardiac College™ (healtheuniversity.ca, accessed on 19 July 2022), which is freely available online in different formats (text, video, and webinars) and in 10 languages.

Social support was associated with support from family members and support from CR providers. Participants suggested that the regular and personalized contact provided by the CR providers motivated them to maintain their exercise routines. As described elsewhere [49], support from family members and peers positively impacts adherence in home-based CR programs and increases up to 35% of the motivation to exercise in overall physical exercise programs [51,52,53]. Interactions and the connection between health care providers and patients were associated with patient engagement in both center-based and virtual CR models [54,55]. Many providers were redeployed during the first wave of the COVID-19 pandemic [26]. As a consequence, patients experienced changes in their CR providers, which may have generated the feeling of losing connection [26]. Specifically for patients with low technology skills and computer literacy, good connection and interaction between providers and patients are crucial to guarantee successful participation in a virtual CR model. Therefore, it is important to increase awareness among CR providers about the importance of their professional endorsement while delivering CR care. It is also important that the same CR provider follows a CR group of participants from the beginning to the end of their participation in the program. That enables the building of stronger connections and trust between patients and providers. In addition, the promotion of virtual CR group sessions is incentivized, as it helps improve peer support in the virtual CR model.

The COVID-19 restrictions implemented to contain the spread of the virus, such as lockdowns, posed several physical and mental barriers for individuals to exercise. In combination with the lack of equipment, access to exercise facilities, social distancing, and the lack of in-person interaction, individuals participating in a virtual CR model also experienced fear to perform outdoor exercise because of their vulnerable condition (i.e., CVD), uncertainty about the future, and difficulties in learning how to exercise without in-person supervision. It is known that engagement in physical activity behaviour can help mitigate negative effects on mental health and enhance individuals’ well-being [56]. The closure of parks, trails, gyms, and CR facilities made it difficult for participants to find spaces to exercise where they feel safe. Thus, the delivery of different possible home exercise programs should be recommended [57,58]. The implementation of virtual exercise group sessions can also improve engagement in virtual CR program models.

Although some participants reported they had more time to exercise due to the COVID-19 restrictions, others identified lack of time as a barrier to physical activity. Lack of time was associated with work and family responsibilities, and is a prevalent barrier described in different populations before and during the pandemic [59,60].

It is important to note that this study was conducted immediately after the rapid implementation of virtual CR, which was a daunting process, full of challenges. Some of the challenges experienced by the CR team included: greater resource requirements, training of staff on virtual access, difficulties with risk stratification and supervision, lack of specific virtual CR delivery standards, staff redeployment, and a lack of experience with virtual care programming by clinical and administrative staff, as well as patients and healthcare providers associated with the care of the patients outside the rehabilitation center [26]. Although most challenges can be overcome and barriers raised by participants can be addressed, virtual care also has the potential to exacerbate existing sociodemographic disparities [59,61]. Therefore, more studies aiming to explore strategies for improving access across wider populations are necessary to include equitable access to comprehensive care.

### Limitations

Caution is warranted when interpreting results of this study. First, this study only included individuals who continued participating in the CR program after the transition from center-based to a virtual CR model due to the COVID-19 pandemic; participants who dropped out were not included in this analysis. Therefore, major barriers to exercise faced by participants who did not have access the virtual CR model were not captured and should form the basis of future research. Second, findings are specific to CR offered at a single outpatient hospital site in Canada; therefore, generalizability is unknown. Third, the physical activity patterns of individuals participating in different CR models of support also need to be explored. 

## 5. Conclusions

Individuals participating in a center-based CR model before the pandemic increased their physical activity patterns over time, while those transitioning from a center-based to a virtual CR model during the pandemic mostly maintained their physical activity patterns. Participants expressed different barriers and facilitators to exercise that may have influenced their exercise patterns. Further investigations into the long-term barriers and facilitators to virtual CR participation are needed.

## Figures and Tables

**Figure 1 jcm-11-04838-f001:**
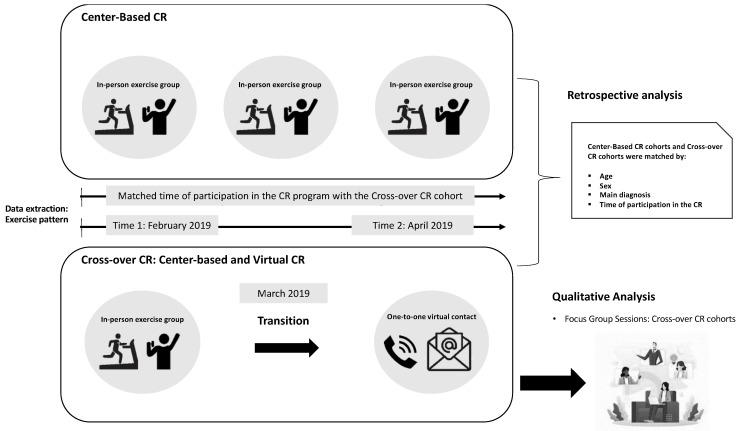
Overview of the sequential explanatory study design. The top box represents the cohort that transitioned from center-based to the virtual CR program due to the COVID-19 pandemic (2020); and the bottom box represents the cohort that initiated and completed participation of the center-based CR program (2019). Exercise pattern extracted in both groups included exercise frequency, duration, and METS/minute. Abbreviation: CR = cardiac rehabilitation.

**Figure 2 jcm-11-04838-f002:**
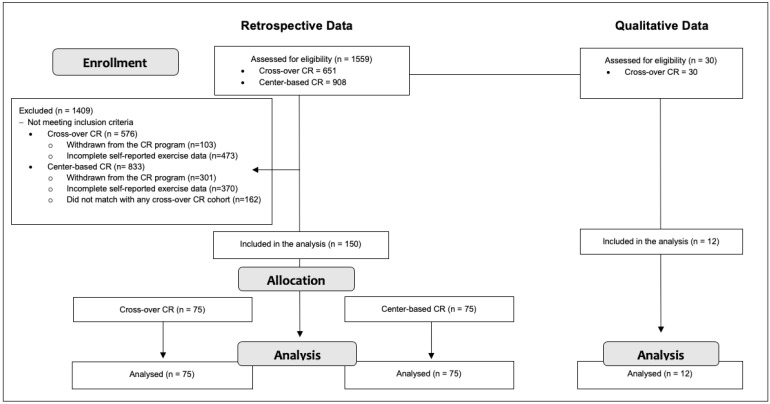
Study flow diagram. Abbreviation: CR = cardiac rehabilitation.

**Figure 3 jcm-11-04838-f003:**
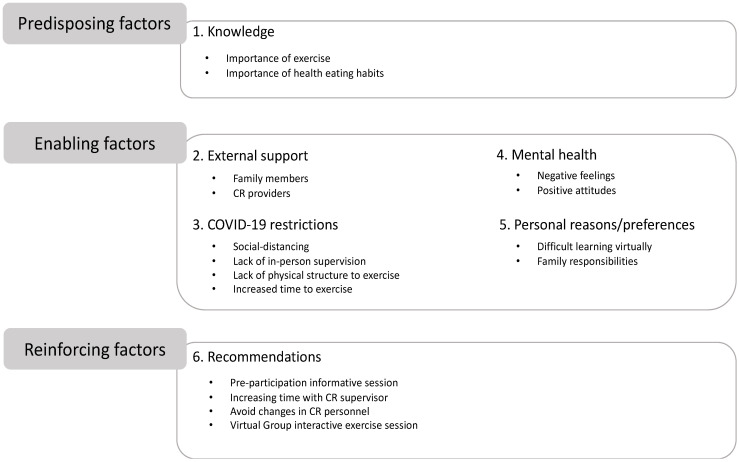
Summary of the focus groups themes and subthemes extracted using the PROCEED-PROCEED model. Abbreviations: COVID-19 = coronavirus 2019; CR = cardiac rehabilitation.

**Table 1 jcm-11-04838-t001:** Characteristics of individuals included in the quantitative analysis at baseline (n = 150).

	Center-Based CR Cohort (n = 75)	Cross-Over CR Cohort (n = 75)	*p* Value
Sociodemographic, and Anthropometric Characteristics			
Sex Male, n (%)	53 (70.6%)	53 (70.6%)	1.000
Age, years (mean ± SD)	67.06 ± 10.01	66.90 ± 10.26	0.909
Time of Participation in a CR program, months (mean ± SD)	2.01 ± 1.05	2.01 ± 1.05	0.916
Body Mass Index, kg/m2 (mean ± SD)	28.35 ± 4.57	26.67 ± 3.59	0.036 *
Referral Diagnosis, and/or Procedure			
Coronary Artery Disease—Coronary Angioplasty, n (%)	44 (58.7%)	44 (58.7%)	1.000
Coronary Artery Disease—CABG, n (%)	13 (17.3%)	13 (17.3%)	1.000
Primary Prevention, n (%)	8 (10.7%)	8 (10.7%)	1.000
Valvular Heart Disease, n (%)	4 (5.3%)	4 (5.3%)	1.000
Stroke, n (%)	3 (4.0%)	3 (4.0%)	1.000
Atrial Fibrillation, n (%)	3 (4.0%)	3 (4.0%)	1.000
Medications in use			
Antiplatelet, n (%)	65 (86.7%)	56 (74.7%)	0.063
Antihypertensive, n (%)	63 (84.0%)	59 (78.6%)	0.221
Lipid-lowering agents, n (%)	63 (84.0%)	65 (86.6%)	0.644
Anticoagulant, n (%)	8 (10.7%)	12 (16.0%)	0.337
Antidiabetic agents, n (%)	7 (9.3%)	20 (26.7%)	0.006 *
Antiarrhythmic, n (%)	1 (1.3%)	1 (1.3%)	1.000

* Difference statistically significant. Abbreviations: CR = cardiac rehabilitation; CABG = coronary artery bypass surgery.

**Table 2 jcm-11-04838-t002:** Comparison of physical activity patterns of individuals who attended a center-based CR program (CBCR, n = 75) and those who transitioned from a center-based to a virtual CR model (cross-over CR, n = 75) during the COVID-19 pandemic.

Variable	CR Program Model	1st Time Point	2nd Time Point	Time *p* Value (n^2^_P_)	Groups (n^2^_P_)	Time vs. Group Interaction (n^2^_P_)
		(Mean ± SD)	(Mean ± SD)			
**Frequency, n times per week**	Cross-over CR cohort	2.7 ± 0.9	2.8 ± 1.9	0.022 (0.04) *	0.079 (0.02)	0.159 (0.014)
	Center-based CR cohort	2.6 ± 1.1	3.1 ± 1.2 *			
**Duration, total minutes/week**	Cross-over CR cohort	124.4 ± 54.6	137.7 ± 63.03 *	0.001 (0.07) *	0.306 (0.007)	0.326 (0.007)
	Center-based CR cohort	138.3 ± 82.3	167.0 ± 95.8 *			
**METS, minutes**	Cross-over CR cohort	426.5 ± 194.4	470.3 ± 229.7	0.001 (0.07) *	0.241 (0.009)	0.510 (0.003)
	Center-based CR cohort	451.6 ± 264.6	551.9 ± 314.7 *			

* Differences comparing to baseline. Bold: difference statistically significant. Time represents the comparison between 1st and 2nd time point. Groups represent cross-over vs. center-based cohort. Abbreviations: CR = Cardiac Rehabilitation; METS = metabolic equivalent; SD = standard deviation; n^2^_P =_ Partial Eta Square.

**Table 3 jcm-11-04838-t003:** Physical activity patterns of individuals who transitioned from a center-based to a virtual CR model (cross-over CR) during the COVID-19 pandemic stratified by sex (female, n = 22 vs. male, n = 53) and age (<65 years old, n = 25 vs. ≥65 years old, n = 49).

Variable	CR Program Model	Age Group	1st Time Point (Mean ± SD)	2nd Time Point (Mean ± SD)	Time *p* Value (n^2^_P_)	Groups (n^2^_P_)	Time vs. Group Interaction (n^2^_P_)
**Frequency, n times per week**	Cross-over CR cohort	<65 years old	2.9 ± 1.6	3.6 ± 1.6 *	0.035 (0.031) *	0.002 (0.095) *	0.030 (0.061) *
		≥65 years old	2.5 ± 0.7	2.8 ± 0.9			
	Center-based CR cohort	<65 years old	3.2 ± 1.1	2.9 ± 1.3			
		≥65 years old	2.4 ± 0.7	2.7 ± 0.9			
**Duration, total minutes per week**	Cross-over CR cohort	<65 years old	124.1 ± 70.4	154.0 ± 85.3	0.028 (0.033) *	0.431 (0.019)	0.931 (0.003)
		≥65 years old	145.3 ± 87.4	173.5 ± 100.7			
	Center-based CR cohort	<65 years old	119.6 ± 57.81	122.9 ± 45.9			
		≥65 years old	126.7 ± 53.3	145.0 ± 69.2 *			
**METS, minutes**	Cross-over CR cohort	<65 years old	409.44 ± 242.25	513.34 ± 290.14	0.072 (0.022)	0.478 (0.017)	0.911 (0.004)
		≥65 years old	472.79 ± 275.06	571.21 ± 327.41			
	Center-based CR cohort	<65 years old	414.56 ± 200.54	429.01 ± 165.72			
		≥65 years old	432.48 ± 193.07	491.02 ± 254.86			
**Variable**	**CR program model**	**Sex**	**1st time point (mean ± SD)**	**2nd time point (mean ± SD)**	**Time *p* value (n^2^_P_)**	**Groups (n^2^_P_)**	**Time vs. group interaction (n^2^_P_)**
**Frequency, n times per week**	Cross-over CR cohort	Female	2.8 ± 1.1	3.1 ± 1.2	0.006 (0.052) *	0.055 (0.052)	0.115 (0.040)
		Male	2.2 ± 0.9	3.0 ± 1.2 *			
	Center-based cohort	Female	2.7 ± 0.8	2.7 ± 1.1			
		Male	2.6 ± 1.1	3.0 ± 1.0			
**Duration, total minutes per week**	Cross-over CR cohort	Female	151.8 ± 87.63	175.3 ± 102.8	0.003 (0.021) *	0.423 (0.019)	0.365 (0.021)
		Male	105.5 ± 57.0	147.1 ± 74.3 *			
	Center-based cohort	Female	125.0 ± 56.5	133.3 ± 56.7			
		Male	122.9 ± 50.9	148.1 ± 76.6 *			
**METS, minutes**	Cross-over CR cohort	Female	489.4 ± 281.2	576.3 ± 333.2	0.015 (0.040) *	0.666 (0.011)	0.559 (0.014)
		Male	360.6 ± 196.5	492.9 ± 262.4 *			
	Center-based cohort	Female	427.2 ± 199.6	457.5 ± 216.2			
		Male	424.8 ± 185.7	501.1 ± 262.4			

* Differences comparing to baseline. Moments represent the comparison between 1st and 2nd time point. Groups represent pandemic vs. center-based cohort. Abbreviations: CR = Cardiac Rehabilitation; METS = metabolic equivalent; SD = standard deviation; n^2^_P =_ Partial Eta Square.

## Data Availability

Data is available upon request.

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
