# Peer review of "Physical Activity Level and Perspectives of Participants Transitioning from Onsite to Virtual Cardiac Rehabilitation during the Early COVID-19 Pandemic: A Mixed-Method Study"

_jcm, 2022, doi:10.3390/jcm11164838_

Round 1
Reviewer 1 Report
Firstly, thank you for opportunity to review very interested article. I don't feel qualified to judge about the English language and style due to not native language.
1. The title reflect the main subject about cardiac rehabilitation during the COVID-19 pandemic, title was clear and easy to understand.
2. The abstract summarize and reflect the work described in the manuscript. However, some of contents was not relevance. I suggest the authors to revised about that.
3. The key words reflect the focus of the manuscript.
4. The manuscript adequately describe the background, present status, and significance of the study. The authors explain situation about COVID-19 and affect in cardiac problems.
5. The manuscript describe methods in adequate detail, study subjects were clear, with demonstrate IRB number or text to human ethics consideration. However, I suggest the authors to clarify in
5.1 Participants to withdrawn during program.
5.2 Participants recorder by..?
6. The research objectives achieved by the experiments used in this study.
7. The manuscript interpret the findings adequately and appropriately, highlighting the key points concisely, clearly, and logically.
8. Tables and figures sufficient, good quality and appropriately illustrative of the paper contents.
9. The manuscript meet the requirements of biostatistics.
10. The manuscript cite appropriately the latest, important, and authoritative references in the introduction and discussion sections. However, some of references were incorrect style for this journal.
Reviewer 2 Report
This original article describes and examines a cardiac rehabilitation program, which was transitioned from face-to-face sessions to distance sessions during the COVID-19 pandemic.
The article addresses a very interesting and actual topic. However, the methodology needs some clarification, and the presentation of the results should also be improved. Please, find my detailed comments below.
Major comments:
Methods part:
- Please explain the circumstances of matching in the retrospective analysis - how did you match patients in the retrospective group and why?
- Please describe the selection of patients in more details in both groups. Explain the flow of figure 2 - why were 739 participants excluded?
- Please provide a more detailed description of interaction analysis you performed - which interactions did you analyse and why? Explain the results of these interactions in the discussion part.
- Focus group sessions and qualitative analysis: please explain why only 12 patients were enrolled into this part of the study. The low number of participants should be included in the limitations. I also would recommend to categorize the answers of the focus group sessions and to summerize them in figures.
Minor comments:
- Figures: the headings of the figures need enhacement for a better understanding. Please, provide a more detailed explanation of figures and explain the abbreviations in the figures, too. (In the same way as you provide it at tables).
- Term “data” is used in plural in my understanding - please correct the verb “was” to “were” after this expression.
Round 2
Reviewer 2 Report
Dear Authors,
Thank you for your responses, I accept all changes you made. Congratulations on your manuscript!